# SLC35A5 Protein—A Golgi Complex Member with Putative Nucleotide Sugar Transport Activity

**DOI:** 10.3390/ijms20020276

**Published:** 2019-01-11

**Authors:** Paulina Sosicka, Bożena Bazan, Dorota Maszczak-Seneczko, Yauhen Shauchuk, Teresa Olczak, Mariusz Olczak

**Affiliations:** 1Faculty of Biotechnology, University of Wroclaw, 14A F. Joliot-Curie St., 50-383 Wroclaw, Poland; psosicka@sbpdiscovery.org (P.S.); Boena.Bazan@gmail.com (B.B.); dorota.maszczak-seneczko@uwr.edu.pl (D.M.-S.); eshauchuk@gmail.com (Y.S.); teresa.olczak@uwr.edu.pl (T.O.); 2Human Genetics Program, Sanford Burnham Prebys Medical Discovery Institute, La Jolla, CA 92037, USA

**Keywords:** nucleotide sugar transporters, Golgi apparatus, glycosylation

## Abstract

Solute carrier family 35 member A5 (SLC35A5) is a member of the SLC35A protein subfamily comprising nucleotide sugar transporters. However, the function of SLC35A5 is yet to be experimentally determined. In this study, we inactivated the *SLC35A5* gene in the HepG2 cell line to study a potential role of this protein in glycosylation. Introduced modification affected neither *N*- nor *O*-glycans. There was also no influence of the gene knock-out on glycolipid synthesis. However, inactivation of the *SLC35A5* gene caused a slight increase in the level of chondroitin sulfate proteoglycans. Moreover, inactivation of the *SLC35A5* gene resulted in the decrease of the uridine diphosphate (UDP)-glucuronic acid, UDP-*N*-acetylglucosamine, and UDP-*N*-acetylgalactosamine Golgi uptake, with no influence on the UDP-galactose transport activity. Further studies demonstrated that SLC35A5 localized exclusively to the Golgi apparatus. Careful insight into the protein sequence revealed that the C-terminus of this protein is extremely acidic and contains distinctive motifs, namely DXEE, DXD, and DXXD. Our studies show that the C-terminus is directed toward the cytosol. We also demonstrated that SLC35A5 formed homomers, as well as heteromers with other members of the SLC35A protein subfamily. In conclusion, the SLC35A5 protein might be a Golgi-resident multiprotein complex member engaged in nucleotide sugar transport.

## 1. Introduction

Nucleotide sugar transporters (NSTs) are transmembrane proteins participating in the glycosylation process. NSTs belong to the solute carrier protein family 35 (SLC35), which now comprises more than 25 members [1]. They localize to the endoplasmic reticulum (ER) and/or Golgi apparatus [2]. The localization pattern of the Golgi-resident NSTs closely follows that of the Golgi-resident glycosylation enzymes, as both types of proteins are concentrated within the central portion of the Golgi stack (i.e., cisternae) but are absent from the Golgi vesicles [3]. A most recent study showed that the guanosine diphosphate (GDP)-fucose transporter encoded by the *SLC35C1* gene is present in the interior and absent from the periphery of the nocodazole-induced Golgi mini-stacks [4]. Interestingly, the most recent data indicate that some members of the SLC35 protein family localize to the endosomes [5,6,7]. Literature data and in silico predictions showed that NSTs consist of an even number of transmembrane helices with both N- and C-termini facing the cytosolic side of the ER/Golgi membrane [8]. According to the working hypothesis, they act as antiporters pumping activated monosaccharides from the cytosol, where their biosynthesis takes place, into the organelle lumen. In parallel, the corresponding monophosphate, an end product of a glycosylation reaction, is transported in the opposite direction [9,10,11,12]. However, it is worth mentioning that such a mechanism was only proven using yeast cells as a model [13].

Until now, mammalian cytidine monophosphate (CMP)-sialic acid (CMP-Sia) transporter (SLC35A1), UDP-galactose (UDP-Gal) transporter (SLC35A2), UDP-*N*-acetylglucosamine (UDP-Glc*N*Ac) transporter (SLC35A3), GDP-fucose (GDP-Fuc) transporter (SLC35C1), and two 3′-phosphoadenosine 5′-phosphosulfate (PAPS) carriers (SLC35B2 and SLC35B3) were extensively characterized. Studies on the first two proteins were more feasible due to the generation of lectin-resistant cell lines with either SLC35A1 or SLC35A2 deficiency. CHO-Lec2 mutant cells do not produce sialylated glycoconjugates due to the mutation in *Slc35a1* gene, resulting in SLC35A1 protein dysfunction [14,15]. The uptake of CMP-Sia into the Golgi lumen of CHO-Lec2 cells is significantly decreased [16]. CHO-Lec8, MDCK-RCA^r^, and Had-1 cell lines are characterized by impaired galactosylation [17,18,19]. In each of these cell lines, the UDP-Gal transporter, encoded by *Slc35a2* gene, is nonfunctional, leading to a drastic decrease of the nucleotide sugar uptake [20]. The UDP-Glc*N*Ac transporter was initially identified in *Kluyveromyces lactis*. It was demonstrated that overexpression of this protein in the *mnn2-2* mutant yeast strain, which produces *N*-glycans lacking terminal *N*-acetylglucosamine, corrected the mutant phenotype [21]. When mammalian homologs of *K. lactis* protein were identified, it was assumed that they act as UDP-Glc*N*Ac carriers. However, it is still unclear if SLC35A3 is the sole transporter of this nucleotide sugar in the Golgi apparatus of mammals. In contrast to the mutant cells deficient in transporters of CMP-Sia and UDP-Gal, cell lines with silenced expression of *SLC35A3* gene synthesize glycans containing *N*-acetylglucosamine. Transport assays indicated that the mutation leads to a 40% decrease of the UDP-Glc*N*Ac uptake into the Golgi vesicles, but the remaining transport seems to be sufficient to maintain proper glycosylation [22]. In opposition to the transporters from the SLC35A protein subfamily, the *SLC35C1* gene encodes a GDP-sugar carrier. Functional identification of the GDP-Fuc transporter was possible due to the discovery of the genetic disorder in humans, which is caused by the *SLC35C1* gene mutation and results in a lack of fucose attached to glycoconjugates [23]. The last group of transporters, which was studied in detail using mammalian cell lines, belongs to the SLC35B protein subfamily. *SLC35B2* and *SLC35B3* genes encode PAPS carriers, and their expression is tissue-specific. Using an RNA interference (RNAi) approach, Kamiyama et al. showed that silencing of these genes resulted in a significant decrease in sulfated glycoconjugates synthesis [24,25,26].

Functions of SLC35B4, SLC35D1 and SLC35D2 mammalian proteins were determined in a heterologous system, based on the nucleotide sugar transport assays performed in the Golgi vesicles isolated from the *Saccharomyces cerevisiae* cells overexpressing respective proteins. These experiments led to the conclusions that SLC35B4 is a UDP-xylose (UDP-Xyl) and UDP-Glc*N*Ac transporter [27], SLC35D1 plays a role as a UDP-glucuronic acid (UDP-GlcA) and UDP-Gal*N*Ac carrier [28], whilst SLC35D2 transports UDP-Glc*N*Ac [29]. However, there is no evidence from either knock-down or knock-out cell lines confirming these findings. The most recently characterized transporter is encoded by *SLC35C2* gene. Overexpression of this gene in CHO cells promoted Notch1 fucosylation, suggesting that SLC35C2 protein might be the second GDP-Fuc carrier [30].

The most recent findings indicate that SLC35 protein family members may play different roles, unrelated to glycosylation. For example, SLC35D3 enhances the formation of protein complexes associated with autophagy [5], SLC35A4 plays an essential role in the subcellular distribution of SLC35A2/SLC35A3 complexes [6], and SLC35F2 exerts an oncogenic effect on papillary thyroid carcinoma progression [31]. However, it is worth noting that these findings come from singular reports, and the solute carrier 35 family members are still primarily considered potential NSTs.

In this study, we aimed to characterize the SLC35A5 protein, which is a potential NST. For this purpose, we employed the HepG2 cell line, characterized by a diversified glycome [32]. We utilized the clustered regularly interspaced short palindromic repeats (CRISPR)/CRISPR-associated 9 (Cas9) approach to inactivate the *SLC35A5* gene and analyzed glycoconjugates produced by the generated knock-out cells. Moreover, we determined SLC35A5’s subcellular localization and C-terminus topology, and we also characterized the interactions of SLC35A5 with other members of the SLC35A protein subfamily. Based on our results, one may not rule out a hypothesis that SLC35A5 might be a member of a Golgi multiprotein complex engaged in nucleotide sugar transport.

## 2. Results

### 2.1. Inactivation of the *SLC35A5* Gene

To functionally characterize the SLC35A5 protein, we employed a CRISPR/Cas9 double nickase approach. For this purpose, HepG2 cells were transfected with appropriate plasmids, and several independent clones were isolated. To confirm that the gene inactivation was effective, genomic DNA and RNA were isolated from HepG2 wild-type and *SLC35A5* knock-out cells. PCR and RT-PCR confirmed that the CRISPR/Cas9 system successfully inactivated the *SLC35A5* gene (Appendix A). In case of clones #1 and #2, PCR products amplified using genomic DNA templates contained a truncated version of the *SLC35A5* sequence, where deletion was not in frame, leading to an early stop codon. In the case of clones #3 and #4, mixtures of non-specific PCR products were obtained. We assumed that deletions in the start region of clones #3 and #4 were large, overlapping one or two of our control primers, making SLC35A5 translation impossible. Among clones verified, only clone #1 contained a mixture of wild-type and mutant cells and, therefore, was not subjected to further analyses (for details, see Appendix A).

### 2.2. Glycosylation Analysis

To examine the putative SLC35A5 role in glycosylation, we characterized glycoconjugates produced by the knock-out cells. Potential changes in *N*- and *O*-glycans were investigated using a panel of 24 lectins (Appendix A). In addition, seven lectins were utilized to characterize glycoconjugates present on the cell surface (Appendix A). Lectin specificity is listed in Appendix A. To analyze oligosaccharides in detail, glycans were isolated and their permethylated 2-aminobenzamide (2-AB) derivatives were analyzed using MALDI-TOF mass spectrometry (Figure 1 and Figure 2). In addition, fluorescently labeled *N*-glycans were separated on a GlycoSep N column and glycosylation patterns were compared (Appendix A). *SLC35A5* gene knock-out did not result in significant changes in *N*- and *O*-glycosylation. Glycolipids extracted from the knock-out cells also remained unchanged (Figure 3). Although, neither heparan sulfate nor keratan sulfate significantly changed upon the introduced genetic modification, a slight increase in chondroitin-4-sulfate level was observed (Figure 4). To verify this finding, we employed an alternative antibody generated toward chondroitin sulfate A. Western blotting analysis confirmed that SLC35A5-deficient cells produced slightly higher levels of chondroitin sulfate proteoglycans as compared to the wild-type cells (Figure 4).

### 2.3. UDP-Sugar Transport Assay

Based on the literature data one may assume that SLC35A5 could be considered a UDP-sugar carrier [33]. Studies performed by Ashikov et al. indicated that SLC35A5 overexpression in *S. cerevisiae* influences neither UDP-Xyl nor UDP-glucose (UDP-Glc) Golgi uptake [27]. Therefore, in this study, we focused only on the four remaining UDP-sugars, namely UDP-Gal, UDP-Glc*N*Ac, UDP-GlcA, and UDP-Gal*N*Ac. We isolated the Golgi vesicle fraction from HepG2 wild-type cells, as well as from *SLC35A5* gene knock-out (clone #2 and #3) cells, and examined the UDP-sugar transport (Figure 5A). SDS-PAGE and Western blotting analyses using four different antibodies specific to Golgi-resident proteins (anti-Golgi matrix protein 130 (GM130), anti-syntaxin 16, anti-mannosyl (α-1,3-)-glycoprotein β-1,2-*N*-acetylglucosaminyltransferase (Mgat1), and anti-SLC35A2), as well as Coomassie Brilliant Blue G250 staining, confirmed equal protein amounts subjected to the transport assay (Figure 5B). Each transport measurement was carried out three times and standardized against UDP-sugar Golgi uptake in the wild-type cells, arbitrarily set as 100%. We demonstrated that the *SLC35A5* gene inactivation did not affect UDP-Gal Golgi uptake. However, the transport of UDP-GlcA decreased by 50% as compared to the wild-type cells. Furthermore, UDP-Glc*N*Ac and UDP-Gal*N*Ac Golgi uptake were reduced in knock-out cells (Figure 5A). Obtained results suggest that SLC35A5 might play a role in nucleotide sugar transport into the Golgi apparatus.

### 2.4. SLC35A5 Subcellular Localization

Our preliminary studies performed on MDCK-RCA^r^ and CHO-Lec8 cell lines indicated that stable overexpression of N-terminally tagged SLC35A5 was not efficient, probably due to protein degradation. We assumed that SLC35A5 might be subjected to a post-translational, physiological process, namely N-terminal proteolytic processing, which could prevent protein detection after N-terminal tagging. However, we were able to successfully overexpress SLC35A5 protein by attaching the fusion peptide at the C-terminus (Appendix A).

*SLC35A5* knock-out cells (clones #2 and #3) were transfected with pSelect-A5-HA plasmid and stable transfectants were generated. After four weeks of selection, cells were subjected to immunofluorescence staining. To detect SLC35A5 protein, mouse anti-hemagglutinin (HA) antibody fluorescently labeled with Alexa Fluor 647 was employed. Endoplasmic reticulum was counterstained with rabbit anti-calnexin primary antibody and goat anti-rabbit IgG Alexa Fluor 568 secondary antibody. To visualize *cis* cisternae of the Golgi apparatus, rabbit anti-GM130 and goat anti-rabbit IgG Alexa Fluor 568 secondary antibody were used. *Trans* cisternae of the Golgi apparatus were detected with rabbit anti-syntaxin 16 primary and goat anti-rabbit IgG Alexa Fluor 568 secondary antibody. As shown in Figure 6, SLC35A5 co-localizes with both *cis* and *trans* Golgi markers and does not co-localize with the ER marker, suggesting that A5 occurs in the Golgi apparatus membrane.

### 2.5. C-Terminus Topology

SLC35A5 has an extremely acidic C-terminus (Appendix A) as compared to other NSTs from the SLC35A protein subfamily. Therefore, we investigated localization of this region using HepG2 cells transiently transfected with pSelect-A5-c-myc or pSelect-Mgat1-HA plasmids (Appendix A). Forty-eight hours after transfection, cells were subjected to immunofluorescence staining with either 0.1% Triton X-100 or 40 µg/mL digitonin as a permeabilization agent. Mgat1, like many other glycosyltransferases, is a type II membrane protein with the C-terminus directed toward the Golgi lumen [34]. Therefore, under such experimental conditions, the HA epitope can only be detected after the membrane permeabilization with 0.1% Triton X-100. In contrast, 40 µg/mL digitonin will only disrupt the plasma membrane. As a positive control of plasma membrane permeabilization, β-tubulin was counterstained.

As expected, anti-HA (Mgat1) staining was only possible after 0.1% Triton X-100 permeabilization. However, anti-c-myc (A5) detection was effective under both conditions (Figure 7A), indicating that the SLC35A5 C-terminus localized toward the cytosol. To additionally verify this result, we overexpressed SLC35A5 also fused with HA epitope (A5/HA) and Mgat1 with c-myc peptide (Mgat1/c-myc). Using transfection and immunofluorescence staining, we confirmed that the C-terminus of SLC35A5 was exposed toward the cytosol (Figure 7B).

### 2.6. Fluorescence Lifetime Imaging Microscopy (FLIM)/Förster Resonance Energy Transfer (FRET) Interaction Analysis

It was demonstrated that NSTs form homomers [35,36,37,38,39,40,41], as well as interact with each other [6,37]. To analyze potential interplay between SLC35A5 and other members of the SLC35A protein subfamily, we used the FLIM/FRET approach. For this purpose, HEK293T cells were employed since the FLIM/FRET interaction analysis protocol was established in our laboratory and optimized for this particular cell line [6,22,37,42]. Cells were transiently transfected with pTag-GFP2-C-A5 plasmid (Appendix A) and enhanced GFP (eGFP) fluorescence lifetime (τ) was measured to establish τ value for the control cells (2.70 ns; Appendix A). Afterward, HEK293T cells were co-transfected with pTag-GFP2-C-A5 and pTag-RFP-C-SLC35A constructs (Appendix A). Both splice variants (A2-Golgi and A2-ER) were tested, although the second splice variant (A2-ER) localized mainly to the ER [43]. However, upon mutual overexpression of SLC35A5 and SLC35A2-ER, a large pool of the UDP-Gal transporter was detected in the Golgi apparatus (Figure 8I). Significant reduction of the eGFP lifetime was observed in case of each protein pair (A5 + A1, 2.29 ns; A5 + A2-Golgi, 2.24 ns; A5 + A2-ER, 2.31 ns; A5 + A3, 2.35 ns; A5 + A4, 2.39 ns; A5 + A5, 2.51 ns; Figure 8; Appendix A). Based on these data one may conclude that SLC35A5 forms homomers and interacts with all other members of the SLC35A protein subfamily.

## 3. Discussion

In this study, we aimed to characterize a yet orphan, putative nucleotide sugar transporter encoded by the *SLC35A5* gene. In order to gain some insight into the role of the SLC35A5 protein in the glycosylation of macromolecules, we inactivated the corresponding gene in HepG2 cells and comparatively examined the most representative classes of glycoconjugates produced by the resulting knock-out cells, i.e., *N*-glycans, *O*-glycans, proteoglycans, and glycolipids. We selected the HepG2 cell line as a model because it produces a wide range of glycoconjugates in relatively large amounts [32]. Surprisingly, we were not able to detect major quantitative or qualitative changes in glycoconjugates synthesized by the SLC35A5-deficient HepG2 cells (Figure 1, Figure 2, Figure 3 and Figure 4). We only observed a slight increase in the level of chondroitin sulfate proteoglycans (Figure 4). Although this effect was confirmed using two different primary antibodies, the impact of the *SLC35A5* gene knock-out on the chondroitin sulfate biosynthesis was rather minor. Importantly, lack of the significant glycosylation changes upon the *SLC35A5* gene inactivation is not an unprecedented phenomenon, since we obtained similar results in the case of the *SLC35A4* knock-out [6]. Moreover, we clearly demonstrated that cells with decreased expression of the *SLC35A3* gene, which encodes the main UDP-Glc*N*Ac transporter, were able to produce glycoconjugates containing significant amounts of Glc*N*Ac [22]. These findings imply that the biological role of proteins from the SLC35 family, as well as the mechanism of their action, is not completely understood. They might also suggest the existence of some as yet unrecognized compensatory capacities of cells.

Importantly, we found a statistically significant decrease in the uptake of UDP-GlcA, UDP-Gal*N*Ac, and UDP-Glc*N*Ac into the Golgi vesicles of SLC35A5-deficient cells, whilst UDP-Gal transport remained unchanged. UDP-GlcA was shown to be transported into the Golgi lumen [44] and our results are in line with this finding. The Golgi pool of UDP-GlcA serves as a substrate for the elongation of chondroitin sulfate and heparan sulfate chains [45]. However, the amount of the corresponding proteoglycans did not decrease upon *SLC35A5* gene knock-out. Strikingly, a number of chondroitin sulfate proteoglycans was even slightly elevated. These findings suggest that cells are able to compensate for the diminished transport of UDP-GlcA, UDP-Gal*N*Ac, and UDP-Glc*N*Ac. The initiation of proteoglycan synthesis, which involves incorporation of GlcA, was also suggested to occur within the Golgi lumen [46]. However, the only so far acknowledged human transporter specific for UDP-GlcA is an ER-resident SLC35D1/UDP-galactose transporter-related 7 (UGTrel7) [28,47]. This protein is also specific for UDP-Gal*N*Ac and, again, no transporters specific for this nucleotide sugar are attributed to the Golgi apparatus.

Our first question was why an impaired import of UDP-GlcA, UDP-Gal*N*Ac, and UDP-Glc*N*Ac into the Golgi lumen had no effect on the glycosylation profile. We provide a number of possible explanations to this apparently surprising phenomenon. There are several ways in which intraluminal pools of these nucleotide sugars could be preserved in case of their diminished uptake by the Golgi vesicles. First of all, Golgi-dependent glycosylation might be supported by the ER-localized NSTs provided that vesicular transport of nucleotide sugars between these two organelles takes place. Such a passive movement of activated sugars was suggested by Kabuss et al. [48] to explain the phenotypic reversion of cells deficient in UDP-Gal import into the Golgi lumen by the ER-localized splice variant of the UDP-Gal transporter. Although such a phenomenon was never actually proven, some evidence suggests that nucleotide sugars are secreted by certain types of cells via exocytosis [49]; thus, it is highly likely that these molecules indeed move forward along the secretory pathway in a largely uncontrolled fashion. In such a case, the influx of nucleotide sugars from the ER to the Golgi lumen could sufficiently compensate for the decreased rate of their transport across the Golgi membrane.

SLC35A5 interacts with SLC35A3, which in turn is specific for UDP-Glc*N*Ac (Figure 8). The idea that SLC35A5 binds many types of nucleotide sugars and passes them to the respective transporters (SLC35A3 in the case of UDP-Glc*N*Ac) may partially explain the decreased Golgi uptake of this nucleotide sugar by SLC35A5-deficient cells. Interestingly, UDP-Glc*N*Ac uptake into the Golgi vesicles of SLC35A3-deficient cells was only decreased by 40% [22]. In light of the current findings, we suggest that this residual uptake might be preserved due to the action of SLC35A5. We also postulate that SLC35A3 and SLC35A5 might enable (either interchangeably or mutually) UDP-Glc*N*Ac transport across the Golgi membrane.

All other members of the SLC35A subfamily (i.e., SLC35A1, SLC35A2, SLC35A3, and SLC35A4) are readily overexpressed as fusion proteins with tags attached at their N-termini [6,20,43,50]. Surprisingly, we were unable to detect N-terminally tagged SLC35A5, while C-tagging was successful. Furthermore, the C-terminally tagged SLC35A5 migrated in SDS-PAGE significantly faster than one can expect based on the theoretical molecular weight of the recombinant protein (Appendix A). These findings allowed us to conclude that SLC35A5 might undergo post-translational, physiological N-terminal proteolytic processing within cells, which appears to be an unprecedented phenomenon in the family of SLC35 proteins. NSTs are either Golgi- or ER-resident proteins. It was shown that the Golgi-resident NSTs, such as the first splice variant of SLC35A2 or SLC35A4, can be tagged either at the N- or C-terminus without affecting their subcellular localization [6,38]. C-terminal tagging only affects subcellular localization of NSTs with C-terminal dilysine motifs [27,43,51], which are considered ER retention/retrieval signals; however, SLC35A5 is not the case. Importantly, most members of the SLC35A subfamily, i.e., SLC35A1, the first splice variants of SLC35A2, SLC35A3, and SLC35A4 localize to the Golgi apparatus [6,43,52]. Therefore, we anticipated that SLC35A5 is also a Golgi-resident NST. Indeed, the recombinant SLC35A5 tagged at the C-terminus with an HA fusion epitope localized exclusively to the Golgi apparatus.

It is widely accepted that NSTs are multitransmembrane proteins with an even number of spans and cytosolic orientation of both N- and C-termini. However, in the case of SLC35A4, some in silico topology predictions suggested an uneven number of spans resulting in luminal orientation of the N-terminus. We firmly demonstrated that the N-terminus of SLC35A4 is directed toward the cytosol and not the Golgi lumen [6]. Here, we also showed that the C-terminus of SLC35A5 is cytosolically oriented (Figure 7). It is worth noting that the C-terminus of SLC35A5 is exceptionally long (over 80 amino acids (aa)) as compared to other members of the SLC35A subfamily [6]. Intriguingly, it is extremely acidic and contains several well-recognized functional motifs, including DXEE (DGEE, aa 404–407), DXXD (DESD, aa 417–420), and DXD (DED, aa 420–422). No other members of the SLC35A subfamily contain such motifs within their C-termini [53,54]. Interestingly, the diacidic DXD motif is highly conserved among many different families of glycosyltransferases and was proposed to indirectly bind nucleotide sugar substrates via coordination of divalent cations [55]. Another acidic motif present within the C-terminal amino acid sequence of SLC35A5, namely DXXD, was shown to be especially efficient in calcium binding by providing a more stable coordinate of this ligand than the DXD motif [56]. Diacidic motifs such as DXE, EXE, EXD, or DXD were repeatedly shown to mediate the export of transmembrane proteins from the ER to the Golgi apparatus [57,58] or cell surface [59]. In the case of yeast Golgi trafficking protein 1 (Sys1p), this phenomenon involves binding of coat protein complex II (COPII) components Sec23p–Sec34p [60]. Interestingly, Sys1p is required for the Golgi targeting of another yeast protein, ADP-ribosylation factor (ARF)-like protein 3 (Arl3p) [61]. The third motif contained within the C-terminus of SLC35A5 (DXEE) was shown to mediate Golgi targeting of the prenylated Rab acceptor 2 (PRA2) [62].

To conclude, here we provide results suggesting that SLC35A5 might be the first mammalian NST engaged in the transport of three different UDP-sugars, i.e., UDP-GlcA, UDP-Gal*N*Ac, and UDP-Glc*N*Ac. Importantly, this was acknowledged from the knock-out-based endogenous uptake assay results and not from the heterologous overexpression systems, as previously done for other putative multi-specific NSTs. Moreover, SLC35A5 is the sole member of the SLC35 family that contains several diacidic motifs within its cytosolically exposed C-terminus, some of which might be capable of UDP-sugar binding. Therefore, we suggest that SLC35A5 could be a superior, regulatory protein that orchestrates the action of other SLC35A subfamily members through heterologous interactions. In addition, our findings suggest that cells have remarkable abilities to compensate for the diminished nucleotide sugar uptake. This study not only adds to our understanding of the mechanism of NST action, but might also demonstrate a previously unrecognized compensatory capacity of cells that enables maintenance of proper glycosylation under conditions of nucleotide sugar depletion.

## 4. Materials and Methods

### 4.1. Construction of Plasmids

Plasmids designed and constructed or used in this study are listed in Appendix A. The cloning strategy and sequences of the guide RNA used for knock-out of the gene are presented in the Appendix A. The mixture of CRISPR/Cas9 double nickase plasmids designed for the human *SLC35A5* gene knock-out, generated by Santa Cruz Biotechnology Inc. (Santa Cruz, CA, USA), is commercially available.

### 4.2. Cell Maintenance and Transfection

For the majority of experiments, HepG2 cells were used. Cells were grown in minimum essential medium Eagle (MEM) supplemented with 10% fetal bovine serum (FBS), 2 mM l-glutamine, 100 U/mL penicillin, and 100 µg/mL streptomycin. For generation of the *SLC35A5* gene knock-out cells, HepG2 wild-type cells were transfected with a mixture of double nickase plasmids according to the manufacturer’s instructions and selected as described previously [6]. For further analyses, four independent clones overexpressing GFP (the second selection marker in the employed CRISPR/Cas9 system) were isolated. To generate the cells overexpressing SLC35A5/HA, *SLC35A5* knock-out cells were transfected with pSelect-A5-HA plasmid using the ESCORT IV transfection reagent (Sigma Aldrich, St. Louis, MO, USA) according to the manufacturer’s instructions. Stable transfectants were generated via selection in MEM complete medium supplemented with 1 µg/mL puromycin and 200 µg/mL Zeocin (InvivoGen, San Diego, CA, USA) for four weeks. To examine SLC35A5 C-terminus topology, HepG2 wild-type cells were transiently co-transfected with either pSelect-A5-HA and pSelect-Mgat1-c-myc or pSelect-A5-c-myc and pSelect-Mgat1-HA plasmids using the ESCORT IV transfection reagent (Sigma Aldrich) according to the manufacturer’s instructions. Cells were seeded onto eight-well microscope slides (Merck, West Point, PA, USA) and analyzed 48 h after transfection. MDCK-RCA^r^ and CHO-Lec8 were grown and transfected as described previously [63] using pVitro-A5-HA plasmid (Appendix A).

FLIM/FRET experiments were carried out using HEK293T cells. Cells were grown in Dulbecco’s modified Eagle’s medium (DMEM) supplemented with 10% fetal bovine serum, 4 mM l-glutamine, 100 U/mL penicillin, and 100 µg/mL streptomycin, transiently transfected with the expression plasmid(s), and seeded as described previously [22]. Cells were analyzed 48 h after transfection in FluoroBrite DMEM medium.

### 4.3. Evaluation of the SLC35A5 Knock-Out Efficiency

Confirmation of the *SLC35A5* gene knock-out was performed at both RNA (RT-PCR) and genomic DNA (PCR) levels. For this purpose, forward 5′-ATGGAAAAACAGTGCTGTAGTC-3′ and reverse 5′-TCTTGTTGAGGAATGCAAGGT-3′ primers for PCR, and forward 5′-ATGGAAAAACAGTGCTGTAGTC-3′ and reverse 5′-CACAGAATGACACAAGCACAC-3′ primers for RT-PCR were used. A detailed procedure was described in our previous report [6], with some modifications. Briefly, amplified genomic DNA fragments were cloned into pJet 1.2 sequencing vector (Thermo Scientific, Waltham, MA, USA). Ten clones obtained from a particular DNA template were sequenced and analyzed, as shown in the Appendix A.

### 4.4. Analysis of Glycoproteins with Lectins

Cell lysates were subjected to SDS-PAGE using 10% polyacrylamide gels and transferred onto nitrocellulose membranes (Whatman, Maidstone, UK). Afterward, glycoconjugates were analyzed using biotinylated lectins as described previously [20]. Lectin specificity is listed in Appendix A.

As an alternative, glycoconjugates present on the cell surface were analyzed using seven different lectins. For that purpose, cells were grown to ~70% confluence, washed with cold Dulbecco’s phosphate-buffered saline (DPBS) buffer, and scrapped. The pellets were washed once more with DPBS, re-suspended in blocking buffer (2% bovine serum albumin (BSA) in DPBS), and incubated on ice for 30 min. Then, the cells were centrifuged, the blocking buffer was removed, and the pellets were incubated with respective lectins diluted in blocking buffer. Succinylated wheat germ agglutinin (sWGA), *Erythrina cristagalli* lectin (ECL), and *Ricinus communis* agglutinin (RCA) were diluted to a final concentration 1 µg/mL, while that of *Aleuria aurantia* (AAL) was 2 µg/mL, and that of peanut agglutinin (PNA), *Maackia amurensis* lectin II (MALII), and *Phaseolus vulgaris* leukoagglutinin (LPHA) was 4 µg/mL. In the case of the negative control, lectin was not added. After 40 min of incubation on ice, the cells were spun down and washed twice with blocking buffer. The cells were fluorescently labeled with streptavidin-Cy3 diluted 1:200 in blocking solution for 30 min, spun down, and washed twice with blocking buffer. Finally, the pellets were re-suspended in 500 µL of blocking solution and kept on ice for fluorescence-activated cell sorting (FACS) analysis. All the steps were performed at 4 °C. After each wash, the cells were spun down for 5 min at 5000× *g* at 4 °C. A NovoCyte Flow Cytometer System was used to analyze the cells. The acquisition parameters were set for a respective negative control. Each experiment was performed in two biological replicates. The data were analyzed with NovoExpress^®^ Software (version 1.2.4).

### 4.5. Analysis of Fluorescently Labeled N-Glycans

*N*-glycans were isolated, purified, and fluorescently labeled with 2-aminobenzamide (2-AB). Labeled derivatives were separated on a GlycoSep N column (Glyco) as described previously [20]. For further characterization of *N*-glycans, permethylation of isolated and 2-AB-labeled oligosaccharides was performed [64]. *N*-glycans were analyzed in positive-ion MALDI-TOF mass spectrometry as described previously [20].

### 4.6. Analysis of Fluorescently Labeled O-Glycans

To characterize *O*-glycans, the procedure reported by Kudelka et al. was adopted [64]. Cells were seeded onto 10-cm culture plates in 10 mL of a complete medium. After 24 h, the complete medium was replaced with 10 mL of medium containing reduced amount of serum (5%), supplemented with 50 µM Ac_3_GalNAcBn. Cells were grown for an additional 72 h; then, the medium was collected and centrifuged (1000× *g*), and the supernatant was subjected to a glycan extraction procedure [64]. Isolated permethylated *O*-glycans were analyzed using MALDI-TOF mass spectrometry.

### 4.7. Isolation and Separation of Glycolipids

Glycolipids were isolated, separated using thin-layer chromatography (TLC), and detected with orcin solution as described previously [6].

### 4.8. Subcellular Fractionation and Nucleotide Sugar Transport Assay

Cells were harvested, washed with 20 mM phosphate-buffered saline (PBS), and suspended in homogenization buffer (10 mM Tris-HCl pH 7.4, 0.14 M KCl, 1 mM MgCl_2_, 0.25 M sucrose). Then, cells were disintegrated using Dounce homogenizer and centrifuged for 1 h at 45,000 rpm at 4 °C. The pellet containing the Golgi vesicle fraction was suspended in 200 µL of the homogenization buffer and protein concentration was determined. UDP-Gal, UDP-Glc*N*Ac, UDP-Gal*N*Ac, and UDP-GlcA transport into the Golgi vesicles was determined as described previously [65] with slight modifications. Reactions were carried out for 10 min at 30 °C in 50-µL samples containing 100 µg of proteins present in the Golgi vesicle fraction, 20 µM cold UDP-sugar (Sigma Aldrich), and 5 µCi tritium-labeled UDP-sugar (American Radiolabeled Chemicals).

### 4.9. Western Blotting

Cell lysates or the Golgi vesicle fraction were subjected to SDS-PAGE using 10% polyacrylamide gels and transferred onto a nitrocellulose membrane as described previously [22]. After transfer, the membrane was blocked for 5 h in 5% (*w*/*v*) non-fat milk PBS/Tween (PBST) solution and then subjected to a reaction with the primary antibody diluted in the blocking solution (Appendix A) overnight at 4 °C. The secondary antibodies were diluted in 1% (*w*/*v*) non-fat milk PBST solution (Appendix A) and applied on the membrane for 1 h. After the primary and secondary antibody incubations, the membrane was washed with 1% (*w*/*v*) non-fat milk PBST solution. For Western blotting employing anti-HA/horseradish peroxidase (HRP) antibody, incubation with the secondary antibody was omitted. Prior to the detection of chondroitin-4-sulfate with MAB2030 antibody (Merck Millipore, Burlington, MA, USA), cell lysates were treated with chondroitinase ABC (Sigma Aldrich) as described previously [6]. Before the heparan sulfate detection, cell lysates were treated with heparinase III (AMSBIO, Abington, UK). The reaction was performed overnight at room temperature in 50-µL samples using 5 µU of the enzyme in 50 mM sodium acetate buffer, pH 7.0, containing 3.5 mM CaCl_2_.

### 4.10. Immunofluorescence

To determine the subcellular localization of the respective proteins, specific antibodies were employed (Appendix A). *SLC35A5* knock-out cells stably overexpressing SLC35A5-HA were washed three times with PBS and fixed for 20 min at room temperature with 4% (*v*/*v*) paraformaldehyde in PBS. Afterward, cells were washed three more times with PBS. Non-specific binding sites were blocked for 1 h with blocking solution containing 1% (*w*/*v*) BSA and 0.1% (*w*/*v*) saponin in PBS. Primary antibodies (Appendix A) were diluted in blocking solution and applied onto the cells for 1 h. Cells were washed three times with blocking buffer. Secondary antibodies (Appendix A) were diluted in blocking solution and applied onto the cells for 1 h. Afterward, cells were washed three more times with blocking buffer. Cell nuclei were counterstained for 10 min with 4′,6-diamidino-2-phenylindole (DAPI; Sigma Aldrich). Then, cells were washed three times with blocking buffer. Microscopy slides were mounted onto glass coverslips using Dako Mounting Medium (Dako, Carpinteria, CA, USA). All performed immunofluorescence experiments were evaluated with a ZEISS LSM 510 confocal microscope.

### 4.11. SLC35A5 C-Terminus Localization

To determine the localization of the SLC35A5 C-terminus, HepG2 cells were transiently co-transfected either with A5/c-myc (encoding SLC35A5) and Mgat1/HA (encoding mannosyl (α-1,3-)-glycoprotein β-1,2-*N*-acetylglucosaminyltransferase/HA) plasmids or with A5/HA and Mgat1/c-myc plasmids. Cells were analyzed 48 h after co-transfection via immunofluorescence as described previously [6], using either 40 µg/mL digitonin (Sigma Aldrich) or 0.1% (*v*/*v*) Triton X-100 as permeabilization reagents. In these experiments, rabbit anti-HA, chicken anti-c-myc, and mouse anti-β-tubulin primary antibodies, as well as goat anti-chicken Alexa Fluor 488, goat anti-rabbit Alexa Fluor 555, and goat anti-mouse Alexa Fluor 633 secondary antibodies, were employed (Appendix A).

### 4.12. Confocal and Fluorescence Lifetime Imaging Microscopy (FLIM)

Confocal and FLIM microscopy were performed as described previously [42]. SLC35A5-eGFP fusion protein was always used as a FRET donor and monomeric red fluorescence protein (mRFP)-tagged transporters from the SLC35A protein subfamily were employed as FRET acceptors.

### 4.13. Bioinformatics

One-way analysis of variance test and Dunnett’s post hoc test were employed to analyze the statistical significance of the results. All statistical analyses were performed using GraphPad Prism software. Transmembrane topology of the SLC35A5 protein was visualized with Protter software [66].

## Figures and Tables

**Figure 1 ijms-20-00276-f001:**
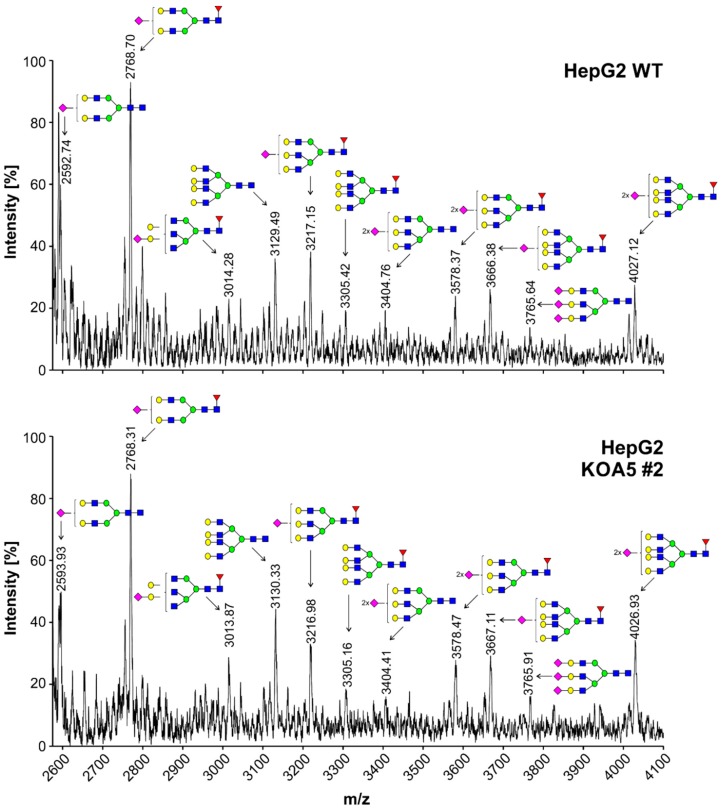
Structural analysis of *N*-glycans isolated from wild-type and solute carrier family 35 member A5 (SLC35A5)-deficient HepG2 cells. The 2-aminobenzamide (2-AB)-labeled and permethylated *N*-glycans derived from the appropriate cells were characterized using MALDI-TOF mass spectrometry. Blue squares, *N*-acetylglucosamine; green circles, mannose; yellow circles, galactose; red triangles, fucose; pink squares, sialic acid.

**Figure 2 ijms-20-00276-f002:**
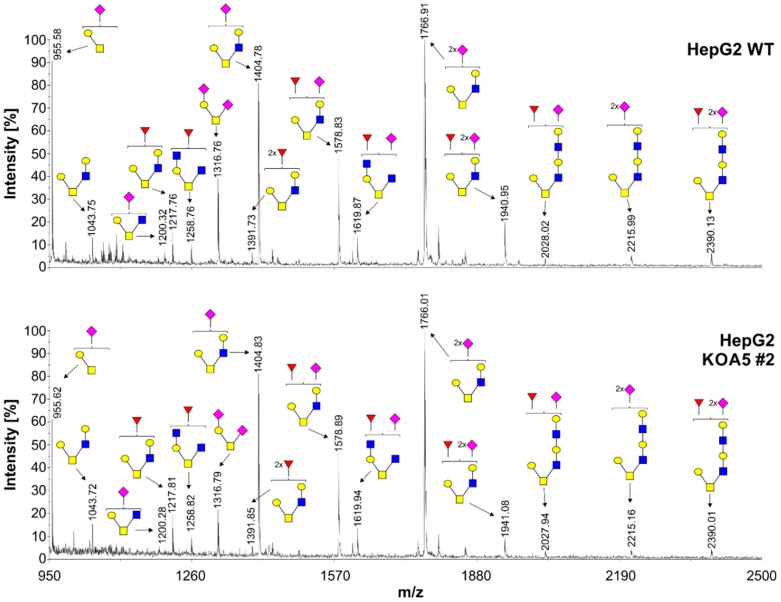
Structural analysis of *O*-glycans isolated from wild-type and SLC35A5-deficient HepG2 cells. Permethylated Bn-*O*-glycans secreted to culture medium were characterized using MALDI-TOF mass spectrometry. Blue squares, *N*-acetylglucosamine; yellow squares, *N*-acetylgalactosamine; yellow circles, galactose; red triangles, fucose; pink squares, sialic acid.

**Figure 3 ijms-20-00276-f003:**
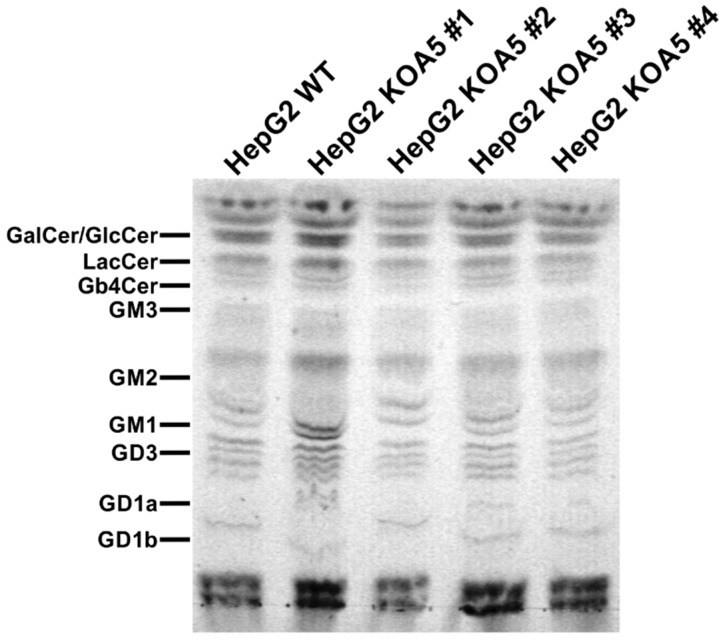
Analysis of glycolipids synthesized by wild-type and SLC35A5-deficient HepG2 cells. Isolated glycolipids were separated using thin-layer chromatography (TLC) and visualized with orcin/H_2_SO_4_ reagent. As a control, a mixture of different glycolipids was also resolved (left line) to ensure that TLC conditions were suitable to separate a diverse range of glycolipids.

**Figure 4 ijms-20-00276-f004:**
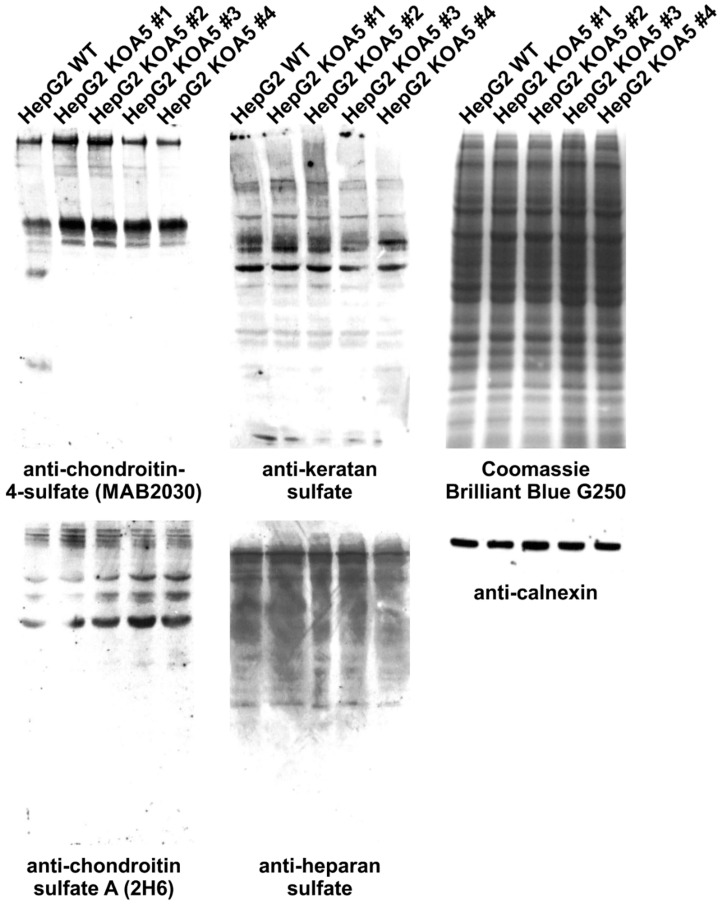
Analysis of proteoglycans synthesized by wild-type and SLC35A5-deficient HepG2 cells. Cell lysates were subjected to SDS-PAGE and Western blotting using specific antibodies raised against proteoglycans. As a loading control, Coomassie Brilliant Blue G250 staining and Western blotting with anti-calnexin antibody were performed.

**Figure 5 ijms-20-00276-f005:**
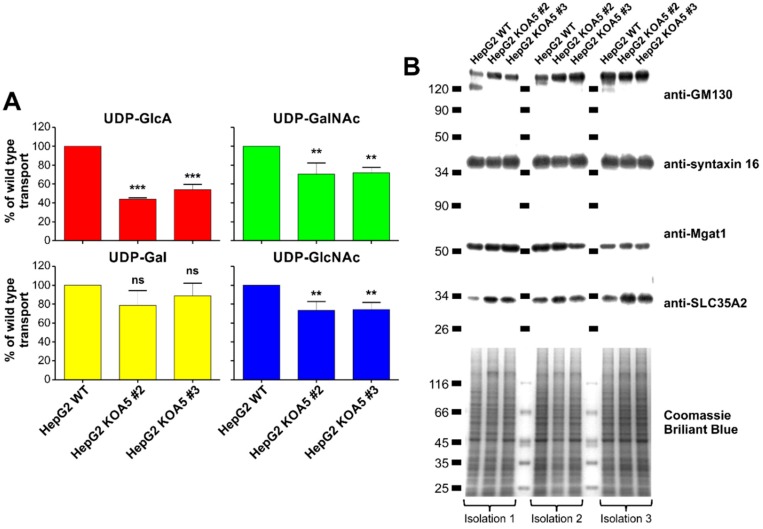
Nucleotide sugar transport assay. (**A**) Uridine diphosphate (UDP)-glucuronic acid (red), UDP-*N*-acetylgalactosamine (green), UDP-galactose (yellow), and UDP-*N*-acetylglucosamine (blue) uptake by the Golgi vesicles derived from wild-type and *SLC35A5* knock-out (clone #2 and #3) HepG2 cells was determined. Data are presented as means ± SD from three independent experiments. ns—no significant difference, *p* > 0.05; ** *p* ≤ 0.01; *** *p* ≤ 0.001. (**B**) To illustrate that the same amount of the protein from the Golgi vesicle fraction was taken for the nucleotide sugar transport measurements, isolated vesicles were separated using SDS-PAGE, and the gel was stained with Coomassie Brilliant Blue G250. Alternatively, four different Golgi proteins were detected using Western blotting.

**Figure 6 ijms-20-00276-f006:**
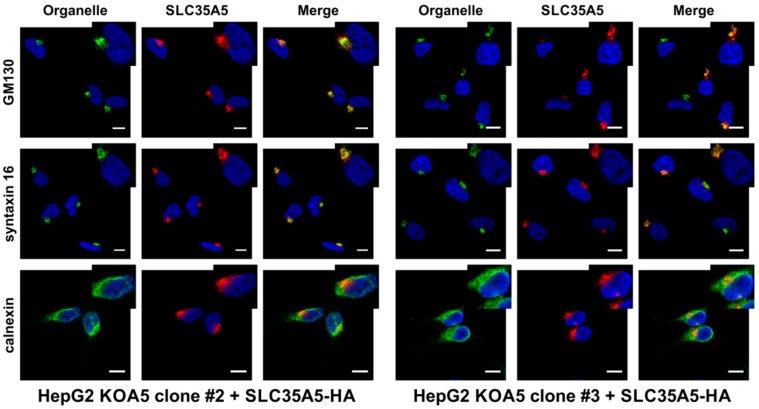
Subcellular localization of SLC35A5 protein. Subcellular localization of SLC35A5 protein was determined in HepG2 *SLC35A5* knock-out cells (clone #2 and #3) stably overexpressing SLC35A5/hemagglutinin (HA). Recombinant SLC35A5 was visualized using mouse anti-HA-Alexa Fluor 647 antibody (red), whilst Golgi and endoplasmic reticulum (ER) were counterstained with specific rabbit antibodies (green). Cell nuclei were counterstained with 4′,6-diamidino-2-phenylindole (DAPI). Scale bar = 10 µm.

**Figure 7 ijms-20-00276-f007:**
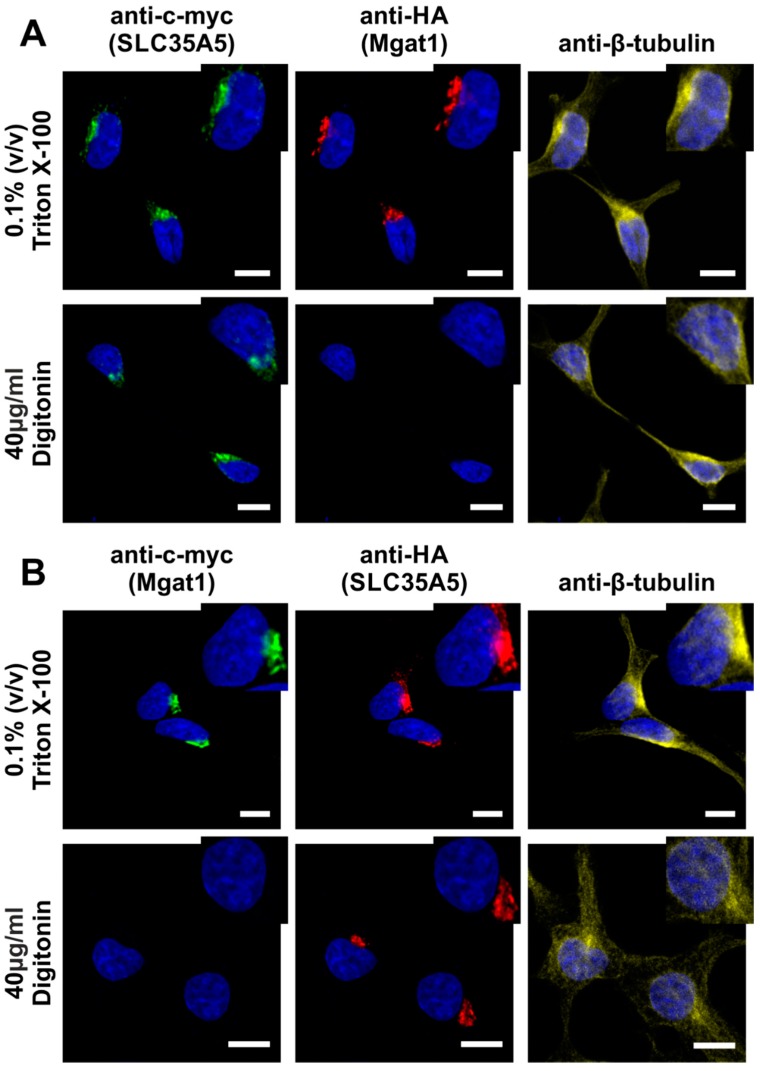
Determination of SLC35A5 C-terminus topology. HepG2 cells simultaneously overexpressing C-terminally tagged SLC35A5 and C-terminally tagged mannosyl (α-1,3-)-glycoprotein β-1,2-*N*-acetylglucosaminyltransferase (Mgat1) were subjected to immunofluorescence staining. (**A**) SLC35A5 was detected with chicken anti-c-myc antibody (green) and Mgat1 was counterstained with rabbit anti-HA antibody (red) or (**B**) SLC35A5 was detected with rabbit anti-HA antibody (red) and Mgat1 was counterstained with chicken anti-c-myc antibody (green). Permeabilization of both plasma and Golgi membranes was performed with 0.1% (*v*/*v*) Triton X-100. To selectively permeabilize the plasma membrane, 40 µg/mL digitonin was employed. As a positive control of plasma membrane permeabilization, β-tubulin was detected using mouse antibody (yellow). Cell nuclei were counterstained with DAPI. Scale bar = 10 µm.

**Figure 8 ijms-20-00276-f008:**
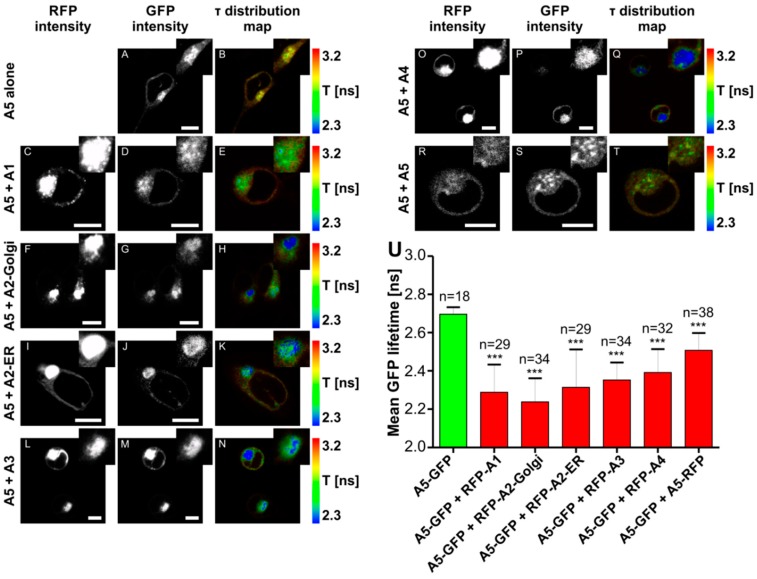
In vivo fluorescence lifetime imaging microscopy (FLIM)/Förster resonance energy transfer (FRET) analysis of interactions between the SLC35A5 protein and other members of the SLC35A subfamily. (**A**–**T**) Confocal intensity-resolved (**A**,**C**,**D**,**F**,**G**,**I**,**J**,**L**,**M**,**O**,**P**,**R**,**S**) and time-resolved (**B**,**E**,**H**,**K**,**N**,**Q**,**T**) images of A5/enhanced GFP (eGFP) (**D**,**G**,**J**,**M**,**P**,**S**) interaction with monomeric red fluorescence protein (mRFP)/SLC35A proteins (**C**,**F**,**I**,**L**,**O**,**R**) in HEK293T cells in comparison with cells expressing A5/eGFP only (**A**). Red-to-blue color shift reflects a shortening of the fluorescence lifetime. The rainbow scale bar placed next to the time-resolved images (**B**,**E**,**H**,**K**,**N**,**Q**,**T**) represents the fluorescence lifetime range between 2.3 (blue) and 3.2 (red) ns. Scale bar = 10 µm; τ—fluorescence lifetime. (**U**) Mean eGFP lifetime values in the absence and in the presence of the acceptor are shown. Data are presented as means ± SD from several measurements of the indicated cell number (*n*). *** *p* ≤ 0.001.

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
