# Peer review of "SLC35A5 Protein—A Golgi Complex Member with Putative Nucleotide Sugar Transport Activity"

_ijms, 2019, doi:10.3390/ijms20020276_

Reviewer 1 Report

The manuscript by Sosicka et al looks into the structure, cellular localization and especially the role of the SLC35A5 protein in a model HepG2 cell line. In particular, the function of this protein as a potential nucleotide sugar transporter and its impact on cell glycosylation (glycoproteins, proteoglycans, glycolipids) are elucidated. The authors have been devoted to the study of other members of the SLC35A protein subfamily in their past research.

The work is sound, well structured and presented. I have detected a couple of minor errors within the text, otherwise the English is at a very good level. I have a strong point concerning the presentation of Supplementary Material though, see below. As sometimes happens, the results are not absolutely clear-cut and in some respects they bring new ambiguities and open new questions. This mainly concerns the discrepancy between significant change in the uptake of some UDP-sugars and no significant influence on the expression of glycostructures (compensation within the cell functions?). The authors discuss possible reasons and give reasonable conclusions. In sum, though the results bring some ambiguous explanations, I consider the work at a good level and suitable for publishing in IJMS, after a minor revision of the following points:

The Supplementary Material should be presented in a single document, starting with a Contents of all the Figures and Tables. Those figures and tables should be named Fig. S1, S2 etc, according to their logical order stemming from the main text, and, importantly, they should all be referenced at appropriate places within the main text, which is not the case.

The cloning strategy and the construct sequences should also be included in the Supplementary Information (or, if not new, appropriately referenced).

The discussion is by way too long and difficult to follow. I recommend to concise it and crop by ca 30%. Possibly, some subtitles may be added if allowed by IJMS.

I recommend to shift one or two less crucial figures in the Supplementary Information to keep the main logical line of the story easier.

Minor points:

- The genera names (K. lactis) should be spelled in full at their first use.
- Please add explanations of abbreviations of lectin names in the legend of Fig2 for better orientation
- I missed the information about the specificity of individual lectin for each glyco-pattern. This should be included (Materials and Methods?)
- Figs. 3, 4 - at least the most important glycans should be identified by names
- Fig. 5 - please add some information on the structures of glycolipids (only acronyms given). One or two explanatory sentences may be included reasoning why exactly these structures were chosen to be monitored.
- The formatting of references does not follow the Instructions for authors,  please correct and add DOI as required by IJMS.

Author Response

Reviewer 1:

Q1

The Supplementary Material should be presented in a single document, starting with a Contents of all the Figures and Tables. Those figures and tables should be named Fig. S1, S2 etc, according to their logical order stemming from the main text, and, importantly, they should all be referenced at appropriate places within the main text, which is not the case.

Response 1

Supplementary materials were improved according the Reviewer’s comments. Supplementary figures and tables are referred in the main text in the proper order.

Q2

The cloning strategy and the construct sequences should also be included in the Supplementary Information (or, if not new, appropriately referenced).

Response 2

Cloning strategy has been described in supplementary materials.

Q3

The discussion is by way too long and difficult to follow. I recommend to concise it and crop by ca 30%. Possibly, some subtitles may be added if allowed by IJMS.

Response 3

The discussion section was shorten (four paragraphs were removed, which resulted in reduction of the text from 2035 words to 1380 words).

Q4

I recommend to shift one or two less crucial figures in the Supplementary Information to keep the main logical line of the story easier.

Response 4

Two figures from the main text were transferred into the supplementary materials.

Q5

The genera names (K. lactis) should be spelled in full at their first use.

Response 5

The style of the name was corrected.

Q6

Please add explanations of abbreviations of lectin names in the legend of Fig2 for better orientation.

I missed the information about the specificity of individual lectin for each glyco-pattern. This should be included (Materials and Methods?)

Response 6

All suitable information regarding lectins names and specificity were provided in supplementary materials (table 2S). Suitable references to supplementary table 2 are indicated in the main text.

Q7

Figs. 3, 4 - at least the most important glycans should be identified by names

Response 7

To our knowledge the only possible name for N-glycans would be, e.g. tetra-antennary complex fucosylated structure, high mannose structure with 7 mannose residues, etc. We believe that such nomenclature does not help in the data interpretation.

In our previous publications (Maszczak-Seneczko et al., 2011, Maszczak-Seneczko et al., 2013, Olczak et al., 2013, Sosicka et al., 2014, Sosicka et al., 2017, Bazan et al., 2018) we used to present MALDI-TOF MS data as it was done in this manuscript, therefore, for clarity reason, we would like to keep the figures shown in the same style.

Q8

Fig. 5 - please add some information on the structures of glycolipids (only acronyms given). One or two explanatory sentences may be included reasoning why exactly these structures were chosen to be monitored.

Response 8

In this experiment, the mixture of different standards was used to optimize the TLC conditions to allow the best separation of different classes of glycolipids (neutral and polar). Because we did not find any differences between HepG2 wild-type and SLC35A5 knock-out cells we did not use more glycolipid standards to identify particular glycolipids present in these cell lines. To clarify this, we added following sentence in the figure legend:

As a control, mixture of different glycolipids was also resolved (left line) to ensure that TLC conditions are suitable to separate diverse range of glycolipids.”

Q9

The formatting of references does not follow the Instructions for authors,  please correct and add DOI as required by IJMS.

Response 9

All references were formatted according to the IJMS instructions.

Reviewer 2 Report

In this paper Paulina et al. try to address the phenotype of the knock out of the putative nucleotide sugar transporter (NSTs) SLC35A5. Using the CRISPR-Cas9 the authors generated several clones, which are mutants for the SLC35A5 gene. They have not could find a significant difference between these mutants clones and the wild type cells in terms of glycoconjugate synthesis, but they show a decrease in UDP-GlcA, -GalNAc and -GlcNAc Golgi uptake in the mutans. They also show that SLC35A5 is localized to the Golgi and it contains several C-terminal acidic motifs, which the authors think may be involved in its transport from the ER to the Golgi.

There are several issues to be resolved before considering this work for publication. The main one is related with the generation of the knock out clones. Since the major point of the paper is getting insights in the function of SLC35A5 looking at the phenotype of null mutants, showing that the work is based on bona fide knock out clones is essential. Thus the authors should make it clear which is the strategy they followed to generate the mutants using CRISPR-Cas9, and why they can identify the mutant just looking at the different size of Genomic and mRNA PCR fragments. At this point I can understand that the authors lack a SLC35 antibody that could show the absence of expression, but they could sequence the different clones to show that they carry mutations in the allele SLC35A5 (insertion or deletion due to the different size of the bands in figure 1) that can prevent the expression of a functional protein (those mutations could results in a open reading frame).

On the other hand there is some inconsistency along the discussion. Unlike the N-terminal tagged protein, they can detect the C-terminal tagged protein, which is localized to the Golgi. In the discussion, they argue they cannot test the putative role of these motifs “since the tag was C-terminally attached, which may cause the motifs of interest to become non-functional”, but at the same time they point out the “Diacidic motifs…might facilitate its (SLC35A5) Golgi targeting”.

Although the authors have not could find effect in glycosylation, they show a defect in UDP-GlcA, -GalNAc, -GlcNAc in Golgi uptake and they try to explain the specific decrease in those nucleotide sugars due to the interaction with other member of the SLC35A family which are involved specifically in the transport of these nucleotide sugars (e.g. SLC35A3). Could the authors identify a lack of interaction of SLC35A with other NSTs, which could explain why the UDP-Gal uptake is not affected?.

 Minor  point:

 In the highlights section, the last statement creates confusion since the C-terminus of SLC35 doesn’t localizes to the cytosol but the C-terminal region of SLC35 is oriented to the cytosol.

Author Response

Reviewer 2:

In this paper Paulina et al. try to address the phenotype of the knock out of the putative nucleotide sugar transporter (NSTs) SLC35A5. Using the CRISPR-Cas9 the authors generated several clones, which are mutants for the SLC35A5 gene. They have not could find a significant difference between these mutants clones and the wild type cells in terms of glycoconjugate synthesis, but they show a decrease in UDP-GlcA, -GalNAc and -GlcNAc Golgi uptake in the mutans. They also show that SLC35A5 is localized to the Golgi and it contains several C-terminal acidic motifs, which the authors think may be involved in its transport from the ER to the Golgi.

Q1

There are several issues to be resolved before considering this work for publication. The main one is related with the generation of the knock out clones. Since the major point of the paper is getting insights in the function of SLC35A5 looking at the phenotype of null mutants, showing that the work is based on bona fide knock out clones is essential. Thus the authors should make it clear which is the strategy they followed to generate the mutants using CRISPR-Cas9, and why they can identify the mutant just looking at the different size of Genomic and mRNA PCR fragments. At this point I can understand that the authors lack a SLC35 antibody that could show the absence of expression, but they could sequence the different clones to show that they carry mutations in the allele SLC35A5 (insertion or deletion due to the different size of the bands in figure 1) that can prevent the expression of a functional protein (those mutations could results in a open reading frame).

Response 1

We agree with the reviewer, but we were not able to find antibody specific to SLC35A5 protein although we tested three different ones. We made an attempt to identify the mutations introduced by CRISPR-Cas9 into genomic DNA. However, in case of clones #3 and #4 we observed a mixture of PCR products which were only detectable at very low levels and we were not able to sequence them. We assume that deletions in the START region were large, overlapping one or two of our control primers (Suppl. Fig1), which caused SLC35A5 translation impossible. In case of clones #1 and #2, amplified PCR and RT-PCR DNAs contained  unspecific products and also truncated versions of SLC35A5 sequence, where deletion was not in frame, leading to early STOP codon.

Q2

On the other hand there is some inconsistency along the discussion. Unlike the N-terminal tagged protein, they can detect the C-terminal tagged protein, which is localized to the Golgi. In the discussion, they argue they cannot test the putative role of these motifs “since the tag was C-terminally attached, which may cause the motifs of interest to become non-functional”, but at the same time they point out the “Diacidic motifs…might facilitate its (SLC35A5) Golgi targeting”.

Response 2

We agree with the reviewer that this part of the discussion was inconsistent. Therefore we removed above-mentioned paragraph from this section. Also few other, most speculative parts of the discussion were removed. Overall the discussion was significantly shorten.

Q3

Although the authors have not could find effect in glycosylation, they show a defect in UDP-GlcA, -GalNAc, -GlcNAc in Golgi uptake and they try to explain the specific decrease in those nucleotide sugars due to the interaction with other member of the SLC35A family which are involved specifically in the transport of these nucleotide sugars (e.g. SLC35A3). Could the authors identify a lack of interaction of SLC35A with other NSTs, which could explain why the UDP-Gal uptake is not affected?.

Response 3

The only known human UDP-galactose transporter is encoded by SLC35A2 gene. We found the interaction between SLC35A5 and SLC35A2 although we did not observe a decrease in UDP-galactose transport in SLC35A5 knock-out cells. It is possibly due to the fact that SLC35A2 has been proven to be the only UDP-galactose transporter. In contrast, SLC35A3 has been demonstrated to contribute to UDP-N-acetylglucosamine transport, however, at least two other UDP-GlcNAc transporters were identified in humans (SLC35B4 and SLC35D2). Therefore we believe that delivery of this nucleotide sugar may be dependent on formation of complexes between SLC35A3 and SLC35A5.

It is worth to point out that our main goal was to demonstrate that the decrease of UDP-GlcA,
-GalNAc and -GlcNAc transport does not correspond with expected glycosylation changes. This is an interesting phenomenon, which was not considered previously. This finding sheds a completely new light on the studies of nucleotide sugar transporters, showing that their role in glycosylation might be much more complex than it was previously assigned.

Minor  point:

Q4

In the highlights section, the last statement creates confusion since the C-terminus of SLC35 doesn’t localizes to the cytosol but the C-terminal region of SLC35 is oriented to the cytosol.

Response 4

This section was indeed misleading and did not fit the IJMS article formatting. therefore, we removed it from the manuscript.

Round  2

Reviewer 2 Report

The author apparently try to address the different issues I had pointed out, and they agree that establishing that they are working whit true knock out clones is essential. I had already indicated in the review that the lack of a proper antibody is  understandable and I suggested they should show the SLC35A5 allele sequence from the mutants clones.

In their reply they say they have amplified by PCR and RT-PCR from clon 1 and 2 the SLC35A5 sequence. Apparently these amplicons contain unespecific products, but the SLC35A5 sequence as well, where they found a deletion that it is not in frame and results in a early stop codon. Nevertheless they don't show that deletion. The sequence file showing the mutation in clone 1 and 2 should be shown. In that way it would even posible to distinguish if the clones are homozygous or heterozygous for that mutation.      

Author Response

Question:

The author apparently try to address the different issues I had pointed out, and they agree that establishing that they are working whit true knock out clones is essential. I had already indicated in the review that the lack of a proper antibody is  understandable and I suggested they should show the SLC35A5 allele sequence from the mutants clones.

In their reply they say they have amplified by PCR and RT-PCR from clon 1 and 2 the SLC35A5 sequence. Apparently these amplicons contain unespecific products, but the SLC35A5 sequence as well, where they found a deletion that it is not in frame and results in a early stop codon. Nevertheless they don't show that deletion. The sequence file showing the mutation in clone 1 and 2 should be shown. In that way it would even posible to distinguish if the clones are homozygous or heterozygous for that mutation.      

Answer:

According to the Reviewer’s suggestion, additional supplementary information was added, which experimentally demonstrates gene deletion. Appropriate changes were also made in the Materials and Methods section.

Materials and Methods

Previous version:

 “Confirmation of the SLC35A5 gene knock-out was performed at both RNA (RT-PCR) and genomic DNA (PCR) levels. For this purpose, forward 5'-ATGGAAAAACAGTGCTGTAGTC-3' and reverse 5'-TCTTGTTGAGGAATGCAAGGT-3' primers for PCR, and forward 5'-ATGGAAAAACAGTGCTGTAGTC-3' and reverse 5'-CACAGAATGACACAAGCACAC-3' primers for RT-PCR were used. Detailed procedure was described in our previous report [6],

 Current version:

“Confirmation of the SLC35A5 gene knock-out was performed at both RNA (RT-PCR) and genomic DNA (PCR) levels. For this purpose, forward 5'-ATGGAAAAACAGTGCTGTAGTC-3' and reverse 5'-TCTTGTTGAGGAATGCAAGGT-3' primers for PCR, and forward 5'-ATGGAAAAACAGTGCTGTAGTC-3' and reverse 5'-CACAGAATGACACAAGCACAC-3' primers for RT-PCR were used. Detailed procedure was described in our previous report [6], with following modifications. Briefly, amplified genomic DNA fragments were cloned into pJet 1.2 sequencing vector (Thermo Scientific). Ten clones obtained from particular DNA template were sequenced and analyzed as shown in supplementary material.”

 Results

 Previous version:

To functionally characterize SLC35A5 protein, we employed CRISPR-Cas9 double nickase approach. For this purpose, HepG2 cells were transfected with appropriate plasmids and several independent clones were isolated. To confirm that the gene inactivation was effective, genomic DNA and RNA were isolated from HepG2 wild type and SLC35A5 knock-out cells. PCR and RT-PCR confirmed that the CRISPR-Cas9 system successfully inactivated SLC35A5 gene (Fig. 1S). Among clones subjected to further analyses, only clone #1 contained a mixture of wild type and mutant cells.

 Current version:

“To functionally characterize SLC35A5 protein, we employed CRISPR-Cas9 double nickase approach. For this purpose, HepG2 cells were transfected with appropriate plasmids and several independent clones were isolated. To confirm that the gene inactivation was effective, genomic DNA and RNA were isolated from HepG2 wild type and SLC35A5 knock-out cells. PCR and RT-PCR confirmed that the CRISPR-Cas9 system successfully inactivated SLC35A5 gene (Fig. 1S). In case of clones #1 and #2, PCR products amplified using genomic DNA templates contained truncated version of the SLC35A5 sequence, where deletion was not in frame, leading to early STOP codon. In case of clones #3 and #4, mixtures of non-specific PCR products were obtained. We assumed that deletion in the START region of clones #3 and #4 were large, overlapping one or two of our control primers, caused SLC35A5 translation impossible. Among clones verified, only clone #1 contained a mixture of wild type and mutant cells and therefore was not subjected for further analyses (for details see supplementary material).”

Round  3

Reviewer 2 Report

The authors have tried to address the issue about the genotype of the different clones and the manuscript should be considered for publication